# Characterization of Extended-Spectrum Beta-Lactamase-Producing *Escherichia coli* in Diarrhoeal Faeces from 0 to 5-Year-Old Children Attending Public Hospitals in Franceville, Gabon

**DOI:** 10.3390/antibiotics13111059

**Published:** 2024-11-07

**Authors:** Anicet-Clotaire Dikoumba, Pierre Philippe Mbehang Nguema, Leresche Even Doneilly Oyaba Yinda, Romeo Wenceslas Lendamba, Jean Constan Obague Mbeang, Guy Roger Ndong Atome, Christophe Roland Zinga Koumba, Sylvain Godreuil, Richard Onanga

**Affiliations:** 1Medical Research and Analysis Unit (URAM), Interdisciplinary Center for Medical Research of Franceville (CIRMF), Franceville BP 769, Gabon; dikoumba@hotmail.com (A.-C.D.); oyabaeven@gmail.com (L.E.D.O.Y.); onangar@yahoo.com (R.O.); 2Omar Bongo Ondimba Army Instruction Hospital (HIAOBO), Libreville BP 20404, Gabon; 3Department of Biology and Animal Ecology, Tropical Ecology Research Institute (IRET), Libreville BP 13354, Gabon; obaguejean@gmail.com (J.C.O.M.); zinga.koumba39@yahoo.com (C.R.Z.K.); 4Département des Opérations Cliniques, Centre de Recherches Médicales de Lambaréné (CERMEL), Lambarene BP 242, Gabon; misterlendamba@gmail.com; 5Department of Chemistry, Faculty of Science, Masuku University of Science and Technology (GABON), Franceville BP 901, Gabon; ndong_atome@yahoo.com; 6Laboratoire de Bactériologie, Centre Hospitalier Universitaire de Montpellier, Université de Montpellier, 34295 Montpellier, France; godreuil@yahoo.fr; 7UMR MIVEGEC IRD-CNRS-Université de Montpellier, IRD, 911 Avenue Agropolis, 34394 Montpellier, France

**Keywords:** multi-resistance, *Escherichia coli*, extended-spectrum beta-lactamase, diarrheagenic *Escherichia coli*, children, public hospitals

## Abstract

Background: In Gabon, studies on the characterization of extended-spectrum beta-lactamase-producing *Escherichia coli* in young children with diarrhoea are almost nonexistent. The objective was to evaluate the prevalence of antibiotic resistance to extended-spectrum beta-lactamase-producing *Escherichia coli* in children at public hospitals in Franceville, Gabon. Methods: Seventy diarrhoea faecal samples were collected from children aged 0–5 years. The culture and isolation of colonies were carried out on MacConkey agar. The colonies were identified using VITEK 2. The determination of the extended-spectrum beta-lactamase’s profiles was accomplished using the double disk method. The identification of phylogroups and pathotypes was performed by PCR. Identification of the ESBL genes was performed by sequencing. Results: A total of 26 strains of *Escherichia coli* (33.0%) were identified from 78 bacterial isolates. Twenty (77.0%) *Escherichia coli* strains carried extended-spectrum beta-lactamases bla_CTX-M-15_ and 5.0% carried bla_SHV-12_ subtypes. Phylogroup D (62.0%) was predominant, followed by B1 (12.0%), B2 (8.0%) and E (4.0%). The bacterial pathogens causing diarrhoea were enterohemorrhagic *E. coli* (12.0%), typical enteropathogenic *Escherichia coli* (8.0%), atypical enteropathogenic *Escherichia coli* (4.0%), Enteroaggregative *Escherichia coli* (4.0%) and enteroinvasive *E. coli* (4.0%). Conclusions: This study showed a high prevalence of extended-spectrum beta-lactamase, *Escherichia coli* of phylogroup D and pathotype enterohemorrhagic *Escherichia coli* in children under 5 years old in public hospitals in Franceville, most probably due to the misuse or inappropriate consumption of beta-lactams.

## 1. Introduction

Diarrhoea is one of the leading causes of infant mortality and morbidity in developing countries [1]. In addition to the classical bacterial pathogens *Shigella*, *Salmonella*, *Yersinia*, *Vibrio* and *Campylobacter* spp., *Escherichia coli* (*E. coli*) strains cause many cases of diarrhoea [2]. Diarrhoea is defined by the World Health Organization as three or more loose or watery stools during a 24 h period [3]. Diarrheagenic *E. coli* (ECD) is one of the most important bacterial agents in the world.

Six intestinal pathotypes of ECD have been described: enterotoxigenic *E. coli* (ETEC), enteroinvasive *E. coli* (EIEC), enteropathogenic *E. coli* (EPEC), enterohemorrhagic *E. coli* (EHEC) or Shiga toxin-producing *E. coli* (STEC), Enteroaggregative *E. coli* (EAEC) and diffuse adherent *E. coli* (DAEC) [4]. Their prevalence and epidemiology may be different depending on the area [4]. The treatment for them has been confronted with the emergence of acquired resistance, such as multi-resistant extended-spectrum beta-lactamase-producing *E. coli* (ESβLE) strains [5], which have a hospital origin (nosocomial infections) due to the systematic prescription of antibiotics by clinicians and patient self-medication [6].

The prevalence of ESβL-producing *E. coli* is somewhat high (45.0% in both Lambaréné and Libreville, Gabon) [7] compared to some central African countries such as Cameroon (12.0%) [8]. Among the ESβLs, CTX-M-type enzymes are most common, and their number has increased rapidly during the last 10 years [9]. The recent global increase has been caused mainly by the bla_CTX-M_-type gene [9]. ESβL-producing isolates, especially CTX-M-producing *E. coli*, exhibit an alarming trend as there is an increasing number of *E. coli* strains with co-resistance to other classes of antibiotics [10].

The worldwide dissemination of ESβL-producing Enterobacteriaceae in human medicine is an urgent problem that poses a serious challenge to the treatment of infectious diseases, particularly the emergence of CTX-M-15-producing *E. coli* [11]. bla_CTX-M-15_ is the most prevalent ESβL gene in human samples worldwide [12]. In addition, bla_CTX-M-15_ and bla_SHV-11_ genes are recognized as plasmid-mediated resistance genes [13].

There are four major phylogenetic groups of *E. coli* strains: A, B1, B2 and D. Commensal *E. coli* are found in groups A and B1, while the extra-intestinal groups are B2 and D [14]. Few studies are available on the association between ECD phylogenetic groups and antibiotic resistance [15].

Specific associations between phylogenetic groups and variants of bla_CTX m_ have been suggested, particularly of bla_CTX-M-14_ and bla_CTX-M-15_, which are considered to be driven by epidemic *E. coli* strains belonging to phylogroup B2 [16].

In children from 0 to 5 years old, researchers do not have enough information on this phenomenon, hence the subject of our study, which aims at assessing the prevalence of antibiotic resistance in the *E. coli* in diarrhoeal patients attending public hospitals in the city of Franceville (Gabon).

## 2. Results

### 2.1. Isolation and Identification of Colonies

The culture and isolation of bacteria from 70 diarrhoeal faecal samples on MacConkey agar plates yielded 78 colonies, of which 58 (74.0%) were pink or red (called lactose-positive) and 20 (26.0%) were colourless or yellow (called lactose-negative). Only lactose-positive strains were considered. Twenty-six (26) strains (out of the 58 lactose-positive) were *E. coli*.

### 2.2. Phylogroups

Of the 26 *E. coli* identified, 22 (85.0%) *E. coli* strains belonged to phylogroups B1, B2, D and E, as follows: 3 (12.0%) B1, 2 (8.0%) B2, 16 (62.0%) D and 1 (4.0%) E. The ECDs were distributed among the phylogroups as follows: 3 (12.0%) EHEC in Group D, 1 (4.0%) atypical EPEC in Group E, 1 (4.0%) typical EPEC in Group D, 1EIEC (4.0%) in Group D and 1 (4.0%) EAEC in Group B2 (Table 1).

### 2.3. Pathogenic E. coli Strains

Of the 26 *E. coli* identified, 8 (31.0%) strains carried pathogenicity genes (ECD), while 18 (69.0%) did not. Among the pathogenic ECD strains, the pathotype EHEC was predominant as it occurred three times (12.0%), followed by two (8.0%) typical EPEC, one (4.0%) EAEC, one (4.0%) EIEC and one (4.0%) atypical EPEC (Table 1).

### 2.4. Antibiotic Susceptibility and Beta-Lactam Resistance Genes

Twenty strains of *E. coli* (77.0%) were positive in the ESβL screening test. For other antibiotics, resistance was reported for kanamycin (90.0%), ciprofloxacin (90.0%), trimethoprim/sulfamethoxazole (50.0%), tobramycin (45.0%), nalidixic acid (45.0%), tetracycline (40.0%), gentamicin (35.0%), netilmicin (30.0%), amikacin (25.0%), fosfomycin (15.0%), chloramphenicol (15.0%), ertapenem (10.0%) and imipenem (5.0%). Our results showed that 19 (95.0%) of the bla_CTX-M-15_ genes were single-carrier and 1 (5.0%) was double-carrier (bla_CTX-M-15_/bla_SHV-12_) (Table 2).

### 2.5. The Prevalence and Phylogroup of MDR ECDs

Among these EsβL-producing *E. coli*, there were eight (31%) of ECD and five (20.0%) of MDR, with all CTX-M-15 belonging to phylogroup D (Table 3).

### 2.6. Some Limitations of the Study

The limitations of our study could be linked to the geographical location of the study town (Franceville), the demographics of this town, and the animist beliefs of the populations:-Geographical limitations: Franceville is about 700 km from the country’s capital (by car). This made it difficult for us to obtain consumables and laboratory reagents.-Demographic limitations: The population of Franceville is very small (around 3%) compared to the population of Gabon. This could explain the small size of our sample.-Animist considerations: this could explain the refusal of some parents to give us faeces samples from their children.

## 3. Discussion

Childhood diarrhoeal diseases have always been a real concern for health institutions worldwide, causing malnutrition and stunting growth. They are one of the principal causes of mortality and morbidity in developing countries. Among the causes of acute bacterial diarrhoea, *Escherichia coli* (*E. coli*) is an important agent of endemic and epidemic diarrhoea across the world [17].

Children aged 0–5 years old interact with their external environment by putting things in their mouths. *E. coli* is one of the most frequently isolated opportunistic pathogens in human and animal faeces [18] and appears to be the leading cause of diarrhoea in children [17]. A study on the aetiology of acute diarrhoea in Tunisian children found a similar prevalence of *E. coli* (76.6%) in the diarrhoeal faeces of children [17].

A study conducted in Ouagadougou (Burkina Faso) between 2013 and 2015 obtained opposing results, with a higher prevalence of EPEC (25.0%) and a similar prevalence of EHEC (12.9%) compared to our results [19]. In another study, the prevalence of EAEC (14.8–23.5%) and EIEC (12.1–22.2%) were higher than EPEC (5.5–13.7%) and EHEC (1.6%) [20]. These trends show that EHEC and EPEC are the main causes of childhood diarrhoea in Tunisia, compared to our results and those of Burkina Faso, where EHEC and EPEC cause more diarrhoea. In developed countries, EPEC, ETEC and EAEC strains are the main causes of infantile diarrhoea [2]. Our results also showed that 18 (69.0%) *E. coli* strains did not carry pathogenicity genes, probably because they were commensal strains [21].

Pathogenic strains of *E. coli* are much more common in phylogenetic group B2 than in group D [22]. Otherwise, phylogroups B2 and D have more virulence properties, such as biofilm formation and haemolysin secretion, compared to isolates from phylogroups A and B1 [23]. Our results showed that the prevalence of phylogenetic group D (62.0%) was higher than that of B1 (12.0%) and B2 (8.0%). Additionally, group D carried pathogenic *E. coli* strains dominated by the EHEC gene. Moreover, among our 18 (69.0%) *E. coli* commensal strains, 11 (42.0%) belong to Group D, 3 (12.0%) to B1 and 1 (4.0%) to B2, compared to another study where commensal *E. coli* were associated with the D (28, 38.0%), A (26, 35.0%) and B2 (12 isolates, 16.0%) phylogroups [15].

In *E. coli* diarrhoeal diseases, treatment is mainly based on the use of third-generation cephalosporins (C3G), aminoglycoside and quinolones [19]. Beta-lactams and fluoroquinolones are the antibiotic classes most commonly used for treating enterobacteria infections in Gabon [24]. Also, aminoglycosides, phenicolates and tetracyclines are in these two classes of antibiotics [25]. Since the emergence of ESβL-producing Enterobacteriaceae, the therapeutic options remain combinations of aminopenicillins and beta-lactamase inhibitors, cephalosporins or fluoroquinolones [7]. However, the use of the appropriate antibiotic is often difficult due to the emergence of resistance [7]. Our study suggested that these *E. coli* strains presented high rates of antibiotic resistance for aminopenicillin, caboxypenicillin, ureopenicillin and monobactam. The lower resistance to carbapenems (imipenem (5.0%) and ertapenem (10.0%)) indicates that these antibiotics are still effective in the treatment of such diarrhoea. Additionally, the prevalence of resistance to kanamycin (96.0%), ciprofloxacin (90.0%), trimethoprim/sulfamethoxazole (50.0%), tobramycin (45.0%), tetracycline (40.0%) and gentamicin (35.0%) was higher than the resistance to fosfomycin (16.0%) and chloramphenicol (12.0%), suggesting that these antibiotics are often used in human clinical therapy in Gabon. These resistance rates are similar to the results recently published in Morocco, where the prevalence of typical antibiotic resistance was 82.0% for ciprofloxacin, 76.0% for trimethoprim/sulfamethoxazole, 66.0% for gentamicin, 56.0% for amikacin and 0.0% for imipenem [7,25]. A study in Lambaréné (Gabon) reported comparable prevalence rates [24]. The low percentages of resistance to carbapenems, fosfomycin and chloramphenicol in this study are like those of other studies, suggesting that these antibiotics are still very active against these pathogens. Also, it has been observed in Morocco that hospitalized patients have a significant prevalence of ESβL-producing *E. coli* (72.0%) [26]. This could be the result of an excessive use of beta-lactams, quinolones and aminoglycosides in this age group in human therapy [24]. Among the ESβLs, CTX-M-type enzymes are most common, and their number has increased rapidly over the last 10 years [9].

The global spread of ESβL-producing Enterobacteriaceae from human clinical strains is a challenge in the treatment of infectious diseases, particularly with the emergence of CTX-M-15-producing *E. coli* [11]. bla_CTX-M-15_ is the most prevalent ESβL gene in human samples worldwide [12]. We found 20 (100%) bla_CTX-M-15_ and 1 (5.0%) bla_CTX-M-15/SHV-12_ (Table 1) recognized as plasmid-mediated resistance genes [24]. Similar studies carried out at Albert Schweitzer Hospital in Lambaréné, Gabon [27], reported the presence of bla_CTX m_ and bla_SHV_ in hospitalized patients. This study confirms that bla_CTX-M-15_ and bla_SHV-12_ are circulating in the Gabonese population, especially in children from 0 to 5 years old.

Previous studies have shown that phylogroups A and B1 (stem commensal) have a high drug resistance profile with few virulence genes, unlike phylogroups B2 and D (pathogenic), which express multiple virulence factors and are more sensitive to antibiotics. In addition, some observations showed that the bla_CTX-M-14_ and bla_CTX-M-15_ genes are associated with phylogroups B2 and A [28]. *E. coli* strains of phylogroup D which were carrying the bla_CTX-M-15_ gene were predominantly of the EHEC pathotype D and are poorly described among children.

## 4. Material and Methods

### 4.1. Ethical Considerations

All patients gave their consent via a signature during the collection period. Anonymous numbers were assigned to participant patients and then transferred to their corresponding samples for confidentiality.

### 4.2. Sampling

Seventy samples of faecal diarrhoea from children aged 0 to 5 years were collected from 22 November 2016 to 29 May 2017, at the Centre hospitalier régional Amissa Bongo (CHRAB) and the Hôpital de coopération sino-gabonaise de Franceville (SGCHF), in the Haut-Ogooué province of southwest Gabon. These samples were taken with the consent of the children’s parents during consultations and at home during certain episodes of diarrhoea.

### 4.3. Culture, Isolation and Identification of Colonies

In the CIRMF Bacteriology Laboratory, each diarrhoeal sample was enriched with tryptone soy broth (BioMérieux, Marcy-l’Étoile, France) and incubated for 24 h. After incubation, 50 μL of each broth was inoculated on a MacConkey agar (MCA) (BioMérieux, Marcy-l’Étoile, France) plate and incubated at 37 °C for 24 h, according to a previous protocol [13]. The bacterial colonies were identified using the VITEK 2 system (BioMérieux, Marcy-l’Étoile, France).

### 4.4. Identification of Pathogenic Strains of E. coli by PCR

For each PCR, DNA was extracted by the boiling method, which was employed on a single colony of each isolate in a final volume of 100 µL of distilled water by incubating it at 95 °C for 10 min followed by a centrifugation step [29]. To identify the pathogenicity genes associated with our 26 *E. coli* strains, two multiplex PCRs were performed, PCR set-1 and PCR set-2. PCR set-1: 5 genes (Eae, pCVD432, LT, STp and STh) were amplified. To obtain a final volume of 50 μL each for 28 tubes, we mixed the following reagents: 25 μL of QIAGEN solution, 10 μL of Q solution, 0.2 μM Eae-f/Eae-r, 0.2 μM pCVD432-f/pCVD432-r, 0.2 μM ST-f/STp-r and 0.2 μM STh-f/STh-r supplemented by 5.2 μL water and 5 μL DNA. PCR set-2: 8 genes (bfp, cadf, ipab, ipah, cada, stx1, stx2 and vtcom) were amplified. To obtain a final volume of 50 μL each for 29 tubes, we mixed the following reagents: 25 μL of QIAGEN solution, 10 μL of Q solution, 0.2 μM Bfp-f/Bfp-r, 0.2 μM CadF-f/CadF-r, 0.2 μM IpaB-f/IpaB-r, 0.2 μM IpaH-f/IpaH-r, 0.2 μM CadA-f/CadA-r, 0.2 μM Stx1-f/Stx1-r, 0.2 μM Stx2-f/Stx2-r, 2.0 μM VTcom-f/VTcom-r supplemented with 6.8 μL of water and 5 μL of DNA. The PCR set-1 and set-2 reactions were performed under the following conditions: denaturation for 15 min at 95 °C and 30 cycles of 30 s at 95 °C, 30 s at 52 °C and 30 s at 72 °C, with a final extension step of 10 min at 72 °C and a hold at 4 °C (Table 4).

### 4.5. Identification of Phylogenetic Groups of E. coli

Phylogenetic group classification (A, B1, B2, C, D, E and F) was performed using the Clermon method by amplification of the 4 genes chua, yjiaa, tspe4c2 and aceK in quadruplex PCR, as described previously [2] (Table 5). DNA extraction was performed via the boiling method.

To obtain a final volume of 25 μL each for 24 tubes, we mixed the following reagents: 2.5 μL of buffer solution (TP), 0.4 mM of dNTPs, 2.5 mM of MgCl2, 0.2 μM of P1/P2, 0.2 μM of P3/P4 and 0.2 μM of P5/P6 supplemented with 11.3 μL of water and 5 μL of DNA. PCR was performed under the following conditions: denaturation for 5 min at 94 °C and 30 cycles of 30 s at 94 °C, 30 s at 59 °C and 30 s at 72 °C, with a final extension step of 7 min at 72 °C and hold at 4 °C.

### 4.6. Antibiotic Susceptibility Testing

Sensitivity studies were conducted according to the Clinical and Laboratory Standards Institute (CLSI) criteria [34]. Antibiotic resistance was assessed via the disk (SirScan Discs, France) diffusion test method on Mueller-Hinton (MH) agar (BioMérieux, Marcy-l’Étoile, France). ESβL production was performed using a double-disk synergy test [30] and confirmed phenotypically on MH agar when the difference in the inhibition zone diameter of one of the cephalosporins (cefotaxime or ceftazidime) alone and combined with a clavulanic acid disc was ≥5 mm.

### 4.7. Identification of ESβL Genes

For the molecular identification of ESβL genes (Table 1), DNA was extracted from a single colony for each isolate into a final volume of 100 μL of distilled water by incubation at 95 °C for 10 min followed by a centrifugation step. The presence of the bla_CTX m_ (CTX m group 1, 2, 8, 9 and 25), bla_TEM_ and bla_SHV_ genes was assessed by multiplex PCR with primers (Table 1) according to a previously published method [35]. DNA from reference bla_CTX_-_M-,_ bla_TEM-_ and bla_SHV-like-positive_ strains was used as a positive control. PCR products were visualized after electrophoresis on 1.5% agarose gels containing ethidium bromide, run at 100 V for 80 min. A 100-bp DNA ladder (Promega, Madison, WI, USA, accessed on 5 June 2017) was used as a size marker. PCR products were purified using the ExoSAP-IT BMC purification kit (GE Healthcare, Piscataway, NJ, USA) [36] and sequenced bi-directionally on a 3100 ABI Prism Genetic Analyzer (Applied Biosystems, Norwalk, CT, USA). Nucleotide sequence alignment and analyses were performed online using the BLAST programme available on the National Center for Biotechnology Information web page http://www.ncbi.nlm.nih.gov (accessed on 14 July 2017). The Statistical Package of Social Science Software 20 was used to determine the frequency of isolated strains and the prevalence of antibiotic resistance.

### 4.8. Accession Numbers

The genome of all genes has been deposited within NCBI/GenBank under the institutional numbers 5(1) bla_CTX-M-15_ [MK590053], 5B(1) bla_CTX-M-15_ [MK590054], 13A(1) bla_CTX-M-15_ [MK590055], 35A bla_CTX-M-15_ [MK590059], 42A(1) bla_CTX-M-15_ [MK590060], 42A bla_CTX-M-15_ [MK590061], 47(1) bla_CTX-M-15_ [MK590062], 54A(1) bla_CTX-M-15_ [ MK590063], 69(3) bla_CTX-M-15_ [MK590064], 90A(1) bla_CTX-M-15_ [MK590065], 92A(1) bla_CTX-M-15_ [MK590066], 10(1) bla_CTX-M-15_ [MK590067], 15(1) bla_CTX-M-15_ [ MK590068], 41A(1) bla_CTX-M-15_ [MK590069], 46A(1) bla_CTX-M-15_ [MK590070], 47(2) bla_CTX-M-15_ [MK590071], 60A(1) bla_CTX-M-15_ [MK590072], 62A(1) bla_CTX-M-15_ [MK590073], 83(1) bla_CTX-M-15_ [MK590074], 90A(2) bla_CTX-M-15_ [MK590075], 69(3) bla_SHV-12_ [MK590076].

## 5. Conclusions

Our data confirmed the circulation of bla_CTX-M-15_ and bla_SHV-12_ in hospitalized infants in Gabon. The results also confirm an emerging resistance to the first-line antibiotics used against bacterial infections in Gabonese hospitals. The alarming fact is that this resistance occurs in opportunistic pathogenic bacteria such as *E. coli*, and mostly its phylogroup D, which known to be virulent and normally susceptible to antibiotics. This alert will enable doctors to improve their antibiotic prescriptions for *E. coli* diarrhoea in small children.

## Figures and Tables

**Table 1 antibiotics-13-01059-t001:** Distribution of pathogenic strains of *E. coli* by pathotype and phylogroups.

				Phylogroups n (%)			
			B1	B2	D	E	N.D.
	EAEC	1 (4)	-	1 (4)	-	-	-
	EIEC	1 (4)	-	-	1 (4)	-	-
ECD n (%)	EHEC	3 (12)	-	-	3 (12)	-	-
	Atypical EPEC	1 (4)	-	-	-	1 (4)	-
	Typical EPEC	2 (8)	-	-	1 (4)	-	1 (4)
Commensal n (%)	*E. coli*		3 (12)	1 (4)	11 (42)	-	3 (12)
Total n (%)			3 (12)	2 (8)	16 (62)	1 (4)	4 (16)

N.D. (Not Determined).

**Table 2 antibiotics-13-01059-t002:** Profiles of ESβL resistance.

Isolate	Antibiotics Profile	ESβL Genes
60A	CIP KAN	bla_CTX-M15_
13A-1	CIP KAN	bla_CTX-M15_
15-1	CIP KAN	bla_CTX-M15_
41A-1	CIP KAN	bla_CTX-M15_
42A-1	CIP KAN	bla_CTX-M15_
46A-1	CIP KAN	bla_CTX-M15_
83-A	CIP KAN	bla_CTX-M15_
54A-1	CIP KAN TET	bla_CTX-M15_
42A	CIP KAN NAL	bla_CTX-M15_
47-1	CIP KAN SXT	bla_CTX-M15_
90A-1	CIP KAN	bla_CTX-M15_
47-2	CHL CIP GEN KAN NAL TOB	bla_CTX-M15_
35A	CIP GEN KAN NAL NET SXT TET TOB	bla_CTX-M15_
5-1	AMK CIP KAN NAL NET SXT TET TOB	bla_CTX-M15_
5B-1	AMK CIP KAN NAL NET SXT TET TOB	bla_CTX-M15_
92A-2	CIP GEN ERT FOS KAN SXT TET TOB	bla_CTX-M15_
62A-1	AMK CIP GEN KAN NAL NET SXT TET TOB	bla_CTX-M15_
69-3	CIP GEN ERT FOS IMP KAN NET SXT TOB	bla_CTX-M15/SHV12_

IMP = imipenem, ERT = ertapenem, GEN = gentamicin, TOB = tobramycin, NET = netilmicin, AMK = amikacin, CHL = chloramphenicol, TET = tetracycline, STX = trimethoprim/sulfonamide, NAL = nalidixic acid, CIP = ciprofloxacin, FOS = fosfomycin, AMP = Ampicillin, KAN = kanamycin.

**Table 3 antibiotics-13-01059-t003:** Prevalence of MDR among diarrhoeal *E. coli* pathotypes.

ECD n (%)	Phylogenic Group n (%)	CIP n (%)	KAN n (%)	NAL n (%)	NET n (%)	SXT n (%)	TET n (%)	MDR n (%)	ESβL Gene
EAEC	B2								
1 (4)	1 (4)	0	0	0	0	0	0	0	blaCTX-M-15
EIEC	D	0	0	0	0	1 (5)	1 (5)	1 (4)	bla_CTX-M-15_
1 (4)	1 (4)								
EHEC	D3 (12)	3 (15)	3 (15)	2 (10)	1 (5)	1 (5)	2 (10)	3 (12)	bla_CTX-M-15_
3 (11.5)									
Atypical EPEC	E1(4)	0	0	0	0	0	0	0	-
1 (4)									
Typical EPEC	D1 (4)	1 (5)	1 (5)	1 (5)	0	0	0	1 (4)	bla_CTX-M-15_
2 (8)									
Total 8 (31)								5 (20)	

GEN = gentamicin, TOB = tobramycin, NET = netilmicin, TET = tetracycline, STX = trimethoprim/sulfonamide, NAL = nalidixic acid, CIP = ciprofloxacin, FOS = fosfomycin, KAN = kanamycin, MDR = Multi-Drug Resistance.

**Table 4 antibiotics-13-01059-t004:** Identification of pathogenic strains of *E. coli* by PCR.

PCR Reaction	Primer ID	Target	Primer Sequences	PCR Product (bp)	References
Quadruplex	chuA.1b	chuA	5′-ATGGTACCGGACGAACCAAC-3′	288	Clermont, O. et al. (2013) [30]
	chuA.2		5′-TGCCGCCAGTACCAAAGACA-3′		Clermont and colleagues (2000) [31]
	yjaA.1b	yjaA	5′-CAAACGTGAAGTGTCAGGAG-3′	211	Clermont, O. et al. (2013) [30]
	yjaA.2b		5′-AATGCGTTCCTCAACCTGTG-3′		Clermont, O. et al. (2013) [30]
	TspE4C2.1b	TspE4.C2	5′-CACTATTCGTAAGGTCATCC-3′	152	Clermont, O. et al. (2013) [30]
	TspE4C2.2b		5′-AGTTTATCGCTGCGGGTCGC-3′		Clermont, O. et al. (2013) [30]
	AceK.f	arpA	5′-AACGCTATTCGCCAGCTTGC-3′	400	Clermont, O. et al. (2013) [30]
	ArpA1.r		5′-TCTCCCCATACCGTACGCTA-3′		Clermont, O. et al. [32]
Group E	ArpAgpE.f	arpA	5′-GATTCCATCTTGTCAAAATATGCC-3′	301	Lescat and colleagues [33]
	ArpAgpE.r		5′-GAAAAGAAAAAGAATTCCCAAGAG-3′		
Group C	trpAgpC.1	trpA	5′-AGTTTTATGCCCAGTGCGAG-3′	219	Lescat and colleagues [33]
	trpAgpC.2		5′-TCTGCGCCGGTCACGCCC-3′		
Internal control	trpBA.f	trpA	5′-CGGCGATAAAGACATCTTCAC-3′	489	Clermont, O. et al. [30]
			5′-GCAACGCGGCCTGGCGGAAG-3′		

**Table 5 antibiotics-13-01059-t005:** Target genes for pathogenic *E. coli* characterization [26].

PCR	Primer	Nucleotide Sequence	Strain	Size
Set-1				
	pCVD432	CTGGCGAAAGACTGTATCAT	EAEC	194 bp
		AAATGTATAGAAATCCGCTGTT		
	LT	ACGGCGTTACTATCCTCTC	ETEC	273 bp
		TGGTCTCGGTCAGATATGTG		
	STp	TCTTTCCCCTCTTTTAGTCAG	ETEC	166 bp
		ACAGGCAGGATTACAACAAAG		
	STh	TTCACCTTTCCCTCAGGATG	ETEC	120 bp
		CTATTCATGCTTTCAGGACCA		
	Eae	TCAATGCAGTTCCGTTATCAGTT	EPEC/EHEC	482 bp
		GTAAAGTCCGTTACCCCAACCTG		
	Bfp	GGAAGTCAAATTCATGGGGGTAT	Typical EPEC	300 bp
		GGAATCAGACGCAGACTGGTAGT		
Set-2				
	Eae	TCAATGCAGTTCCGTTATCAGTT	EPEC/EHEC	482 bp
		GTAAAGTCCGTTACCCCAACCTG		
	CadA	TTCAAAAACATCGATAACGA	EIEC	680 bp
		ACGGTATGCACCGTGAAT		
	Stx-1	CAGTTAATGTGGTGGCGAAGG	EHEC	384 bp
		CACCAGACAATGTAACCGCTG		
	Stx-2	ATCCTATTCCCGGGAGTTTACG	EHEC	584 bp
		GCGTCATCGTATACACAGGAGC		
	VTcom	GAGCGAAATAATTTATATGTG	EHEC	518 bp
		TGATGATGGCAATTCAGTAT		

## Data Availability

Data are contained within the article.

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
