# Peer review of "Characterization of Extended-Spectrum Beta-Lactamase-Producing Escherichia coli in Diarrhoeal Faeces from 0 to 5-Year-Old Children Attending Public Hospitals in Franceville, Gabon"

_antibiotics, 2024, doi:10.3390/antibiotics13111059_

Round 1

Reviewer 1 Report

Comments and Suggestions for Authors

1. It is advisable not to use abbreviations in the title or in the abstract.

2. In the conclusion (abstract), the results are generalized to public hospitals in  in Franceville. However, the study design carried out does not allow such a statement to be made. Conclusions should be limited to the hospital and study period.

3. In the case of stool cultures, what was the reason for incorporating susceptibility to Gentamicyin, Tobramycin, Netilmicin, Fosfomycin and Kanamycn.

4. What criteria were used to classify the strains as MDR?

5. Why were beta-lactam antibiotics not included in the evaluation of the sensitivity profile?

6. It would be advisable for the authors to comment on the limitations of the study.

7. The wording of the document must be improved. The scientific names are poorly written, just as the resistance genes detected are not written according to the established criteria.

Author Response

  1. It is advisable not to use abbreviations in the title or in the abstract.

The title revised is

 Characterization of Extended-spectrum betalactamase-producing Escherichia coli in diarrhoeal faeces from 0 to 5-year-old children attending public hospitals in Franceville, Gabon

  1. In the conclusion (abstract), the results are generalized to public hospital in Franceville. However, the study design carried out does not allow such a statement to be made. Conclusions should be limited to the hospital and study period.

Franceville has two public hospitals. We collected our faeces samples in these two hospitals. So the results obtained concern these two public hospitals.

  1. In the case of stool cultures, what was the reason for incorporating susceptibility to Gentamicyin, Tobramycin, Netilmicin, Fosfomycin and Kanamycn.

These antibiotics are recommanded by Clinical and Laboratory Standards Institute (CLSI) guidline to be test in Gram negative bacteria like E. coli. Also Gentamicyin, Tobramycin, Netilmicin and Kanamycn are antibiotic from Aminoglycosides familly wich are one of the antibiotics familly mainly used in the treatment of gastroenteritis caused by E. coli.

  1. What criteria were used to classify the strains as MDR?

The term MDR applies mainly to S. aureus, Enterococcus spp, Enterobacteriaceae (other than Salmonella and Shigella), P. aeruginosa and Acinetobacter spp, due to their epidemiological importance, the emergence of antimicrobial resistance and the importance of these bacteria in the healthcare system.

In literal terms, MDR means ‘resistant to more than one antimicrobial agent'. But The definition most frequently used for Gram-positive and Gram-negative] bacteria is ‘resistant to three or more antimicrobial classes'.

  1. Why were beta-lactam antibiotics not included in the evaluation of the sensitivity profile?

In our previous study on antibiotic resistance in fruit bats, published in the journal Microorganism (Mbehang et al, 2020), we showed that when a Gram-negative bacillus produces a betalactamase (E. coli), it is resistant to aminopenillins (Amoxicillin, Ampicillin, Amoxicillin+Clavulanic acid), carboxypenicillin (Ticarcillin, Temocillin), ureidopenicillin (Piperacillin, Piperacillin+Tazobactam), monobactams (Aztreonam), and first-, second- and third-generation cephalosporins. This is why, in the present work, we wanted to find out whether Betalactamase-producing E. coli could be resistant to other families of antibiotics.

  1. It would be advisable for the authors to comment on the limitations of the study. Done after the conclusion

    Limitations of the study: We have had difficulty to obtain faecal samples from the children in Franceville area. Franceville is the capital of the second province of Gabon. It is located approximately 700 km from the country's capital (by car). This semi-urban environment allows that the populations of this area use both traditional and modern medicine. Therefore, the hospital is the last resort for sick people. Very few parents took their children with diarrhea to the hospital, but some parents refused to participate in the study. Also, people do not fully understand the importance of scientific research. Some even thought that we want to take their children's feces for fetish practices. The above could explain the low quantity of fecal samples obtained in 5 months of sampling.

    We also had great difficulty obtaining the laboratory reagents needed to analyze the fecal samples. Delivery times for ordered products ranged from 3 to 4 months from our suppliers. This caused us to loose certain products upon delivery.

  2. The wording of the document must be improved. The scientific names are poorly written, just as the resistance genes detected are not written according to the established criteria.

These adjustments have been made

Reviewer 2 Report

Comments and Suggestions for Authors

Well described study with some interesting results but some improvement is necessary:

1. The names of the bacteria should be in italic.

2. Line 48 - It should be [4].

3. Line 77 - State the full name of "MC".

4. Line 79 - "Only lactose + colonies were considered." This part of the study is lacking as all colonies must have been tested for E. coli and identified by other methods because some E. coli strains form lactose - negative colonies.

5. Lines 206-209 and 219-222 - The information is duplicated.

6. Line 209 - "25 E. coli strains" - Aren't they 26?

Author Response

1. The names of the bacteria should be in italic: Done

2. Line 48 - It should be [4]. Done

3. Line 77 - State the full name of "MC". Done

4. Line 79 - "Only lactose + colonies were considered." This part of the study is lacking as all colonies must have been tested for E. coli and identified by other methods because some E. coli strains form lactose - negative colonies.

Thank you very much for your comments.

5. Lines 206-209 and 219-222 - The information is duplicated.

Lines 206-209 have been removed.

6. Line 209 - "25 E. coli strains" - Aren't they 26? There are 26 strains of E. coli

Reviewer 3 Report

Comments and Suggestions for Authors

In the attachment

Author Response

  1. Bacterial names should be in italics e.g. lines 2, 22, 23, 28, 29, 39, 40, 42, 45 etc. The Authors should check it carefully. Done
  2. Acronym ES L should be introduced and explained in the same place in the text: Done
  3. Write ES L instead of ESBL : Done
  4. Line 49 multi-resistant - what does multidrug-resistant mean? In literal terms, MDR means ‘resistant to more than one antimicrobial agent'. But The definition most frequently used for Gram-positive and Gram-negative] bacteria is ‘resistant to three or more antimicrobial classes'
  5. Line 48 area4 what does it mean?: It was a mistake. It's about [4]. We have changed it in the manuscript.
  6. Line 56, 63 etc. The names of the genes should be written blaCTX-M, blaSHV: Done
  7. Line 77 what are MC plate mean? Add explanation : This was MacConkey Agar (MCA) instead of MC. We have changed it in the manuscript.
  8. Lines 78-79 lactose-positive instead of lactose+ : Done
  9. Line 78 the % should be written 74.0%, 26.0% etc., first place after coma: Done
  10. Line 80 change colonies to cultures or strains: Done
  11. Line 89 what is pathovar? Did you mean pathotype? We have replaced pathovar with pathotype
  12. Table 2 should be isolate not isolat : Done
  13. Line 116 developing countries1? What does it mean: It was developing countries, not developing countries1 . we have corrected it in the manuscript
  14. Line 202 add country : Done
  15. Line 241 add company : Done
  16. Antimicrobial susceptibility testing. First step should be double-disc synergy test, second step confirmation, cephalosporin alone and with clavulanic acid. Add write citations: Jarlier, Appleton. Thank you for your comments.
  17. Line 119 “Under 0” What does it mean?: We changed “Children under 0 to 5 years of age interact “ by “Children aged 0 to 5 years interact”.
  18. Line 239 we have 2024 year. Why criteria from 2017? We carried out our faecal sample collection in 2017.
  19. Explain acronym CLSI: Clinical and Laboratory Standards Institute
  20. Limitations of the study should be added: Done

Limitations:

We have had difficulty to obtain faecal samples from the children in Franceville area. Franceville is the capital of the second province of Gabon. It is located approximately 700 km from the country's capital (by car). This semi-urban environment allows that the populations of this area use both traditional and modern medicine. Therefore, the hospital is the last resort for sick people. Very few parents took their children with diarrhea to the hospital, but some parents refused to participate in the study. Also, people do not fully understand the importance of scientific research. Some even thought that we want to take their children's feces for fetish practices. The above could explain the low quantity of fecal samples obtained in 5 months of sampling.

We also had great difficulty obtaining the laboratory reagents needed to analyze the fecal samples. Delivery times for ordered products ranged from 3 to 4 months from our suppli-ers. This caused us to lose certain products upon delivery.